# Cytotoxic and Genotoxic Effects of Composite Resins on Cultured Human Gingival Fibroblasts

**DOI:** 10.3390/ma14185225

**Published:** 2021-09-11

**Authors:** Francesco De Angelis, Domitilla Mandatori, Valeria Schiavone, Francesco Paolo Melito, Silvia Valentinuzzi, Mirco Vadini, Pamela Di Tomo, Lorenzo Vanini, Letizia Pelusi, Caterina Pipino, Piero Del Boccio, Camillo D’Arcangelo, Assunta Pandolfi

**Affiliations:** 1Department of Medical, Oral and Biotechnological Sciences, University “G. d’Annunzio” of Chieti-Pescara, 66100 Chieti, Italy; fda580@gmail.com (F.D.A.); domitilla.mandatori@unich.it (D.M.); _valeria_95@libero.it (V.S.); francescomelito.97@gmail.com (F.P.M.); m.vadini@unich.it (M.V.); pamela.ditomo@unich.it (P.D.T.); vaniniodonto@gmail.com (L.V.); letizia.pelusi@unich.it (L.P.); caterina.pipino@unich.it (C.P.); camillo.darcangelo@unich.it (C.D.); 2Center for Advanced Studies and Technology—CAST (ex CeSI-MeT), University “G. d’Annunzio” of Chieti-Pescara, 66100 Chieti, Italy; silvia.valentinuzzi@unich.it (S.V.); piero.delboccio@unich.it (P.D.B.); 3Department of Pharmacy, University “G. d’Annunzio” of Chieti-Pescara, 66100 Chieti, Italy

**Keywords:** dental resins, composite resins, Bis-GMA, TEGDMA, gingival fibroblast, cytotoxicity, biocompatibility

## Abstract

The aim of the study was to evaluate the cytotoxic and genotoxic potential of five commercially available dental composite resins (CRs), investigating the effect of their quantifiable bisphenol-A-glycidyl-methacrylate (Bis-GMA) and/or triethylene glycol dimethacrylate (TEGDMA) release. Experiments were performed using the method of soaking extracts, which were derived from the immersion of the following CRs in the culture medium: Clearfil-Majesty-ES-2, GrandioSO, and Enamel-plus-HRi (Bis-GMA-based); Enamel-BioFunction and VenusDiamond (Bis-GMA-free). Human Gingival Fibroblasts (hGDFs) were employed as the cellular model to mimic in vitro the oral cavity *milieu*, where CRs simultaneously release various components. Cell metabolic activity, oxidative stress, and genotoxicity were used as cellular outcomes. Results showed that only VenusDiamond and Enamel-plus-HRi significantly affected the hGDF cell metabolic activity. In accordance with this, although no CR-derived extract induced a significantly detectable oxidative stress, only VenusDiamond and Enamel-plus-HRi induced significant genotoxicity. Our findings showed, for the CRs employed, a cytotoxic and genotoxic potential that did not seem to depend only on the actual Bis-GMA or TEGDMA content. Enamel-BioFunction appeared optimal in terms of cytotoxicity, and similar findings were observed for Clearfil-Majesty-ES-2 despite their different Bis-GMA/TEGDMA release patterns. This suggested that simply excluding one specific monomer from the CR formulation might not steadily turn out as a successful approach for improving their biocompatibility.

## 1. Introduction

Dental restorative materials based on composite resins (CRs) are commonly used to perform adhesive restorations, with direct or indirect techniques. Being bonded to tooth structures, they allow for the preservation of sound dental tissues and represent an excellent aesthetic solution which can easily be repaired even after many years. However, following their widespread diffusion, increasing doubts have been raised over the last years about their safety [1,2].

Biocompatibility is the ability of a material to induce an appropriate biological response following a specific application [3]; it is a complex and dynamic process that can change over time due to the interaction between the host environment (the patient), the material, and the function that it should perform. Under this perspective, almost every material that is placed in the oral cavity cannot be considered as inert; when in contact with a tissue, they may interact and a biological response may be generated. In this context, several in vitro studies have shown that some components included in CRs can be potentially cytotoxic [1,4,5,6,7]. In particular, CRs are composed of an organic polymeric matrix, fillers (inorganic particles such as crystalline quartz, pyrogenic silica, glasses of barium, zinc and strontium, ceramic), bonding agents, pigments, catalysts, and inhibitors [8]. Organic components present in the matrix, above all bisphenol-A-glycidyl-methacrylate (Bis-GMA) derived monomers, but also triethylene glycol dimethacrylate (TEGDMA), urethane dimethacrylate (UDMA), and 2-hydroxyethyl methacrylate (HEMA), seem able to affect the biological compatibility [2,9,10,11]. Indeed, although methacrylate compounds are commonly employed to improve clinically relevant CR features such as viscosity, flexural strength, water sorption/solubility, and volumetric shrinkage [12], several adverse effects have been associated with their use. The potential cytotoxicity of the CR organic components is mainly due to the residues of free methacrylate monomers following the phase of polymerization, which may trigger the production of prostaglandin E2 (PGE2), the expression of cyclooxygenase 2 (COX2), and a pro-inflammatory activation through the increase of interleukin-1β (IL-1β), IL-6, and nitric oxide (NO) [13,14,15]. Direct CR restorations are light-cured at body temperature, while indirect restorations are subjected to supplementary heat-curing cycles, and both seem to undergo an additional post-curing process at body temperature after restoration placement [16]. The final degree of conversion value has been reported to widely range between 50% and 90%, 24 h after light-curing [17]. This incomplete reaction leads to the release of monomers into the oral cavity during a variable period [18]. Subsequently, in later stages, biocompatibility can be further influenced by other factors such as erosion, degradation, and the presence of bacteria at the interface between restoration and dental tissues [11,18,19,20,21,22,23].

Beyond the biological effect of resin monomers, great attention has also recently been focused on the cytotoxic potential of nano-fillers within nanofilled/nanohybrid CRs [24,25,26,27,28,29,30], which seem to be dependent on particle size, surface area and structure, chemical composition, solubility, shape, and aggregation. Small agglomerates or the non-agglomerated form may easily penetrate physiological barriers and travel within the circulatory systems of a host [31]; in an agglomerate state, nanoparticles appear less dangerous.

As a response to the potential hazards due to cytotoxic free monomers and/or non-agglomerated nanoparticles, numerous efforts have been made by manufacturers over the last years to provide alternative solutions that could reduce CR cytotoxicity. One possible approach could be the exclusion of some well-known cytotoxic components from the CR formulation; Bis-GMA-free composites have also been made commercially available for several years. Nevertheless, the real effect on living cells of every new CR formulation that is placed on the market is not always completely foreseeable and would definitely benefit from any possible additional investigation.

To date, most of the studies on dental composite cytotoxicity and genotoxicity have dealt with the effects of single CR components [32,33,34]. However, less information is available about cytotoxicity, ROS release, or DNA damages induced by composite extracts (or soaking media), which consist of multiple components. Indeed, extract-based experiments might represent a better option than single-component experiments, since they mimic a situation that more closely resembles the oral milieu [2].

Based on these, the aim of this in vitro study was to evaluate, by means of the dental composite extract approach, the potential cytotoxic effects of five different CRs on hGDFs, while investigating the effects of the Bis-GMA and/or TEGDMA content. Experiments were carried out taking into account cell metabolic activity, reactive oxygen species (ROS) production, and genotoxicity (phosphorylated-Gamma-H2A histone, γH2AX cell expression) outcomes.

## 2. Materials and Methods

### 2.1. Realization of Composite Disks and Soaking Extracts

The list of the five CRs (Bis-GMA based or free) selected for the study is given in Table 1. The cytotoxic and genotoxic effects of CRs on hGDFs were tested using the soaking-derived extracts, which were produced according to the International Organization for Standardization (ISO; no. 10993-5:2009).

Cylindrical composite disks were manufactured for each CR, having a total surface of 50.27 mm^2^ (diameter 4 mm, height 2 mm), positioning the uncured material in different polyvinylsiloxane molds. To avoid contamination, a different mold was used for every different type of resin. Then, molds were inserted between two glass slides and stuck with a paper clip for 20 s to extrude the excess material.

The disks were light-cured for 40 s from the upper surface using a lamp (Celalux 3, VOCO, Cuxhaven, Germany) with an 8 mm diameter tip and an output power of 1300 mW/cm^2^, and were kept dry for 24 h, sterilized under a hood, by UV radiation, for 15 min on each side [35].

For every different composite, 2 CR disks with a 50.27 mm^2^ surface were placed in a plastic tube with 1 mL of culture medium composed of Dulbecco’s Modified Eagle Medium (DMEM) low glucose (cat. D6046, Sigma-Aldrich, Saint Louis, MO, USA), supplemented with 10% fetal bovine serum (FBS), 100 mM glutamine (L-Glu), and 1% penicillin/streptomycin (P/S) (CTRL). This led to a ratio between the surface area of exposed composite disks and the volume of the culture medium equal to 100.54 mm^2^/mL. CR disks were kept immersed in the control medium for 24 h and 14 days (CR soaking extracts) at 37 °C in static conditions (Figure 1).

### 2.2. Soaking Extract Preparation for LC–MS/MS Analysis

The actual presence or not of Bis-GMA and TEGDMA in the soaking extracts was investigated through LC–MS/MS analysis, which required the following procedures. A total of 200 µL of soaking samples underwent protein precipitation with 3 volumes (600 µL) of cold methanol following incubation on ice for 30 min; samples were centrifuged for 30 min at 13,000× *g* (4 °C), and supernatants were evaporated to dryness under nitrogen.

Finally, residues were reconstituted with 2 volumes (400 µL) of acetonitrile:water (ACN:H_2_0) 30:70 (v/v) containing 0.1% formic acid (FA). For quantitative purposes, a calibration curve was assessed, preparing calibration standards in a culture medium DMEM supplemented with 10% FBS, 100 mM L-Glu, and 1% P/S, with known concentrations of TEGDMA and Bis-GMA provided by Sigma-Aldrich (linear range 0.8–100 ng/mL, r^2^ = 0.997 for TEGDMA; linear range 3–800 ng/mL, r^2^ = 0.984 for Bis-GMA). Samples were analyzed in triplicates as they were for the TEGDMA determination and diluted 1:10 for Bis-GMA quantification; hence, standard concentrations covered the entire range of possible concentrations encountered during the analysis. The analytical blank consisted of the cell culture media without the presence of composite materials. The limit of quantification (LOQ) was 0.8 ng/mL for TEGDMA and 3 ng/mL for Bis-GMA.

### 2.3. LC–MS/MS Analysis

LC–MS/MS analysis was carried out following the work of Polydorou et al. [36] and using an Alliance High Throughput (HT) 2795 HPLC system (Waters Corp., Milford, MA, USA) coupled with a Micromass Quattro Ultima Pt mass spectrometer (Waters Corp., Milford, MA, USA). Five µL of samples were injected at a flow rate kept at 250 µL/minute, and analytes were separated on a Phenomenex Luna 3μ C8 (2) 100 Å HPLC column (50 × 4.6 mm) (Phenomenex, Castel Maggiore, Italy) with a SecurityGuard™ cartridge (Phenomenex, Castel Maggiore, Italy). The binary gradient consisted of (A) H_2_O and (B) ACN, both containing 0.1% FA, using the following conditions: 30% B, isocratic for 1 min; linear increase to 95% B within 6 min; linear increase to 99% B within 2 min, kept for 1 min; return to the initial condition within 6 min and kept for 5 min. Mass Spectrometry (MS) parameters used to acquire in multiple reaction monitoring (MRM) mode are listed in Table 2, in terms of cone potential, collision energy, and parent/daughter ion transitions. Data acquisition was performed in positive electrospray ionization mode (ESI+), with a total run time of 21 min, injection-to-injection. Raw data were processed by QuanLynx™ (Waters Corp., Milford, MA, USA). Representative MS/MS spectrum and extracted ion chromatogram (EIC) for both the analytes are shown in Appendix A.

### 2.4. Cell Culture

The cytotoxicity of the five CRs was assessed on hGDFs isolated from human gingival biopsies obtained from seven patients undergoing partial gingivectomy procedures in the dental clinic of the University G. D’Annunzio Chieti-Pescara (CE, N° 1968-24/07/2020). Briefly, gingival specimens were subjected to an enzymatic digestion for 1 h at 37 °C using a solution of collagenase type 1A (Sigma Aldrich, Saint Louis, MO, USA) and dispase (Sigma Aldrich, Saint Louis, MO, USA).

Then, the residual gingival samples were placed in a petri dish with the culture medium (CTRL) to favor a final spontaneous migration of the cells. The isolated hGDFs were grown in a controlled atmosphere (5% CO_2_ and 37 °C) upon reaching the confluence and were used for all experiments between the 3° and the 6° passage.

The characterization of hGDFs was performed by evaluating the expression of the following markers: CD105 (FITC-conjugated antibody; Becton Dickinsons BD Bioscience, cat.326-040), CD73 (PE-conjugated antibody; Becton Dickinsons BD Bioscienc, cat.550257), CD90 (FITC-conjugated antibody; Becton Dickinsons BD Bioscience, cat.555595), CD326 (PerCP-Cy5.5-conjugated antibody; Becton Dickinsons BD Bioscience, cat.347199), and CD45 (FITC-conjugated antibody; Becton Dickinsons BD Bioscience, cat.196-040). FACSVerse (BD Bioscences, San Jose, CA, USA), FACSDiva v 6.1.3, IDEAS software (BD Biosciences, San Jose, CA, USA), and FlowJo 8.3.3 software (Tree Star Inc, Ashland, OR, USA) were used for the cytometric analysis.

For cellular experiments, hGDFs were treated for 48 h with the culture medium (CTRL condition) or with the soaking extracts derived from the five selected CRs, obtained at two different times of soaking (24 h and 14 days), as previously reported. Soaking extracts were used at three different concentrations (100%, 50%, and 25%) (Figure 1).

### 2.5. 3-(4, 5-dimethylthiazolyl-2-yl)-2, 5-diphenyltetrazolium Bromide (MTT) Cell Viability Analysis

The effect of the CR soaking extracts on hGDF metabolic activity was assessed by performing the MTT assay (cat. M211281G, Sigma Aldrich), according to the manufacturer’s instructions. 

Cells were seeded in a 96-well plate at the density of 5000 cells/well (20,000 cells/mL) and were incubated with the culture medium (CTRL) or CR extracts (100%, 50%, and 25%) obtained at soaking times of 24 h and 14 days. After 48 h of treatment, 20 µL of MTT solution (5 mg/mL) were added to each well and incubated for 3 h at 37 °C. Then, 200 µL of Dimethyl sulfoxide (DMSO) were added (30 min at 37 °C) and the spectrophotometric reading was carried out at a wavelength of 540 nm using a microplate absorbance reader (SpectraMAX 190, Molecular Devices, San Jose, CA, USA).

### 2.6. Reactive Oxygen Species (ROS) Flow Cytometry Analysis

ROS released by hGDFs cells (20,000 cells/mL) were measured following the treatment with the culture medium (CTRL) or the CR extracts (100%, 50%, and 25%) obtained at soaking times of 24 h and 14 days. The hydrogen peroxide (H_2_O_2_, 300 µM for 30 min) was used as positive control. After 48 h of exposure, cells were collected and stained using the cell fluorescent reagent “CellROXTM Green Reagent” (C10444, Molecular Probes, ThermoFisher Scientific, MA, USA) at 2.5 µM in PBS for 30 min at 37 °C. Each sample was processed using a FACS Canto II flow cytometer (BD Bioscences, San Jose, CA, USA). All data were analyzed using FACSDiva v 6.1.3, IDEAS software (BD Biosciences) and FlowJo 8.3.3 software (Tree Star Inc., Ashland, OR, USA). The results were obtained as an MFI (Mean Fluorescence Intensity) Ratio calculated by dividing the MFI of positive events by the MFI of negative events (MFI of secondary antibody).

### 2.7. γH2AX Evaluation by Immunofluorescence

Immunofluorescence analysis was used to evaluate DNA damage induced in the hGDFs treated with the culture medium (CTRL) or the undiluted CR-derived extracts obtained at soaking times of 24 h and 14 days. Doxorubicine (DOXO, 1µM for 30 min) was used as a positive control. The level of γH2AX was detected by immunostaining with γH2AX (Ser139) primary antibody (1:800, Cell Signaling Technology, Danvers, MA, USA) and Alexa Fluor 488 anti-rabbit secondary antibody (cat. A11034, Invitrogen, ThermoFisher Scientific, MA, USA). Immunofluorescence experiments were performed on hGDFs seeded on sterile glass cover slips (12 mm diameter) in 24-well plates (20,000 cells/well; 20,000 cells/mL). Following 48 h of treatment, cells were fixed in 4% Paraformaldehyde (10 min RT), permeabilized with Triton (0.1%; 10 min RT), stained with anti-γH2AX primary antibody (cat. 2577, Cell Signaling Technology; 1:800, overnight 4 °C) and with Alexa Fluor 488 anti-rabbit secondary antibody (cat. A11034, Invitrogen; 1:1000, 1 h RT). 4′, 6-diamidino-2-phenylindole (DAPI; cat. D9542, Sigma-Aldrich; 1:1000, 15 min RT) was used to stain the nuclei and to observe the cells through a confocal microscope (Zeiss LSM-800, Carl Zeiss Meditec AG, Oberkochen, Germany). Data, calculated as γH2AX percentage (%) of positive cells, were obtained by analyzing at least three different fields for each image with ImageJ software (NIH, US, ImageJ software).

### 2.8. Statistical Analysis

Data were expressed as mean ± standard error (SEM). Statistical analysis was performed using the Kruskal–Wallis and the Dunn’s post hoc tests. The Shapiro–Wilk normality test was used to verify the normal distribution of the data. The α value was set at 0.05. Analyses and graphs were performed using GraphPad Prism Software Analysis (version 6).

## 3. Results

### 3.1. Quantification of TEGDMA and Bis-GMA from Soaking Extracts by LC–MS/MS Analysis

Results from the quantification of TEGDMA and Bis-GMA by LC–MS/MS analysis on soaking extracts obtained by the immersion of CRs in culture media (24 h and 14 days; scheme in Figure 1) are shown in Figure 2.

As reported in Figure 2a, observable levels of TEGDMA were found only after 24 h of exposure to VD and VOCO resins (1.2 ng/mL and 7.9 ng/mL, respectively).

On the other hand, the quantification of Bis-GMA revealed a detectable concentration of this monomer, not only at the first time point but also after 14 days of immersion (Figure 2b). More specifically, Bis-GMA levels were found to be higher than 70 ng/mL and 800 ng/mL in all the samples after 24 h and 14 days of exposure, respectively. A general upward trend, following the soaking time, is appreciable, except for samples exposed to the VOCO composite; concerning UE and ES-2, the growing measure of Bis-GMA reached dramatically high concentrations by µg/mL orders of magnitude. Interestingly, the difference of the Bis-GMA levels at the two soaking times was found to be statistically significant for the VD (*p*-value: 0.0003), ES-2 (*p*-value: 0.02), and BF-2 (*p*-value < 0.0001) resins.

### 3.2. Effects of CRs on hGDFs Viability

Before setting up the cell metabolic activity experiments, hGDFs were firstly characterized for their phenotype. As shown in Figure 3a, cells obtained from gingival biopsies through our combined isolation method based on enzymatic digestion and spontaneous migration exhibited the characteristic fibroblast-like morphology and expressed the typical markers CD105, CD73, CD90, but not CD45 and the epithelial marker CD326 (Figure 3b).

Successively, based on the experimental plan reported in Figure 1, hGDFs were treated for 48 h with the two soaking times for extracts (24 h and 14 days) derived from each type of CR.

As shown in Figure 4, the treatment with all three concentrations (100%, 50%, and 25%) of the BF-2, VOCO, and ES-2 extracts (24 h and 14 days: Figure 4a,b respectively) did not induce significant effects on cells metabolic activity. On the other hand, the Bis-GMA free resin VD and the Bis-GMA-based UE significantly reduced cell numbers compared to the basal condition (CTRL) at both soaking times (Figure 4).

### 3.3. Effects of Dental Resins on the hGDF Cell ROS Production

The ROS cytometric analysis revealed that the BF-2, VOCO, and ES-2-derived extracts induced a slight increase in oxidative stress, which did not reach statistical significance compared to basal conditions (CTRL; at both times of soaking used (24 h and 14 days; Figure 5a,b respectively).

As regards the VD and UE resins, the low number of metabolically active cells induced by the treatment with undiluted extracts did not allow ROS quantification in both experimental conditions. However, the exposure to their dilution (50% and 25%) reduced the hGDF mortality and, notably, the ROS levels, at soaking times of 14 days in both cases; levels comparable to those induced by the positive control H_2_O_2_ were also reached (Figure 5b).

### 3.4. Effects of Dental Resins on Phosphorylated γH2AX

In order to better investigate the effects of CRs, a possible DNA damage induced in hGDFs cells was investigated through the evaluation of the γH2AX expression. Worthy of note is the fact that compared to CTRL (basal condition), the treatment with the undiluted extracts of almost every CR increased the percentage of positive cells for γH2AX (Figure 6a,b).

DOXO was employed as positive control, and the percentage of positive cells for γH2AX in cell cultures treated for 48 h with VOCO (24 h extracts) and ES-2 (24 h and 14 day extracts) was significantly lower compared to DOXO condition (Figure 6a,b). The Bis-GMA free resin VD increased the percentage of positive cells for γH2AX at both times of soaking, while, as expected, the resin UE showed significance with the 14-day soaking extract treatment.

All these results were confirmed by representative immunofluorescence images showed in Figure 6a,b.

## 4. Discussion

Cytotoxicity evaluation is one of the essential tests used to assess the biocompatibility of materials that have to be employed on human beings. Nowadays, despite the undiscussed popularity of dental CRs, there are increasing concerns about the potential cytotoxicity and genotoxicity of the components that these materials may release [2,9,10,11,18,19,20,21,22,23].

In relation to this, Bis-GMA and other methacrylate monomers such as TEGDMA are the most commonly employed in composite material fabrication since they allow for the positive adjustment of clinically relevant features such as viscosity, flexural strength, water sorption/solubility, and volumetric shrinkage [12]. However, a wide range of adverse effects have been associated with their use. In particular, several in vitro studies demonstrated that Bis-GMA may stimulate the production of PGE2, COX2 expression, and the pro-inflammatory activation of IL-1β, IL-6, and NO [13,14,15]. As a consequence, the attempt to switch towards Bis-GMA-free composite materials has become of particular interest for manufacturers in order to minimize the cytotoxic potential of their products.

For this reason, in the present study, after having assessed the quantifiable methacrylate Bis-GMA and TEGDMA residual monomers, the effects of the soaking extracts obtained from five commercially available dental CRs (Table 1) were tested in terms of cytotoxicity, ROS production, and DNA damages. To this aim, a cellular model of primary isolated human gingival fibroblasts (hGDFs; Figure 1) was used. In particular, the cells were not placed in direct contact with the CRs but were treated with the soaking extracts obtained from the immersion of dental CR disks in the culture medium for 24 h and 14 days. Indeed, this experimental method simulated a condition closer to the oral milieu [2], where the oral soft tissues could be negatively affected by the residual monomers derived from the various CR components following saliva interaction [37]; this resulted in the most commonly used approach in the literature at the moment [38,39].

First of all, although the CRs selected for the study presented various components (Table 1) in their formulation that could influence the CR properties, we confirmed by LC–MS/MS analysis of our soaking extracts the actual residual (or not) of the most common methacrylate monomers, with potential cytotoxicity employed in dental resin formulations such as Bis-GMA and TEGDMA.

Our results showed that observable levels of TEGDMA monomers were found only after 24 h of exposure to VD and VOCO resins. The higher amount of residual TEGDMA monomer found within the first 24 h is in accordance with the evidence that small molecular weight monomers such as TEGDMA have higher mobility and polarity that enable them to be released faster than other large molecules; thus, TEGDMA-based CRs can release a high quantity of monomers into aqueous environments, as already reported by Moharamzadeh K et al. [40]. The same authors demonstrated that TEGDMA monomers disappear after seven days of incubation of TEGDMA-based CRs in culture media due to their enzymatic degradation and aggregation, mainly with albumin. Our data agree with this phenomenon, showing undetectable levels of TEGDMA monomers after 14 days of exposure (Figure 2a). Based on these observations, we can speculate that enzymatic degradation and protein aggregation occurring in culture media over time could lead to a decrease in the measurable TEGDMA monomer by our LC–MS/MS method. Therefore, more in-depth kinetic studies of this phenomenon are needed to better evaluate the cytotoxicity of different TEGDMA isoforms and derivatives. In this context, further proof is provided by several other studies demonstrating that an enzyme-catalyzed hydrolysis of CRs can occur by esterases, as those typically present in inflammatory sites and saliva. Accordingly, the complexity and enzymatic heterogeneity of serum-addicted culture media can mimic the rate of hydrolysis in methacrylate-based polymers, as shown by Finer and Santerre in terms of decreasing the amount of the residual TEGDMA monomer by esterases when compared to PBS soaking [41,42]. On the other hand, regarding the increasing levels of Bis-GMA, there is a general consensus that esterases do not show the same specificity for monomers and that CR degradation relies on the specific enzyme dose response [43].

Actually, Bis-GMA monomers were found at noticeable concentrations at both 24 h and 14 days of exposure, in almost all the investigated CRs (Figure 2b). These data confirmed the presence of such a compound in the composition of the investigated resins and are in partial agreement with their specific manufactures’ formulation reported in Table 1. Indeed, although BF-2 and VD CRs were indicated as Bis-GMA-free by their material composition declaration, through LC–MS/MS, low levels of Bis-GMA monomers were measured in both the resins. These results are consistent with data recently published by Šimková M and colleagues [44], and in agreement with this, it is important to point out that this study also showed the release percentage of Bis-GMA from Bis-GMA-free resins to be very low compared to the average release of other resins. Indeed, by contrast, the resins containing Bis-GMA (VOCO, UE, and ES-2) were associated with the release of high levels of monomers in the culture media, reaching concentrations of µg/mL orders of magnitude, which might be considered clinically relevant as their magnitude is close to the maximum levels of Bis-GMA clinically quantified in human saliva by Michelsen et al. (2.149 µg/mL) [45,46].

On the other hand, the residual percentage of Bis-GMA from Bis-GMA-free resins (BF-2 and VD) compared to the average release of others (considered 100%) is about 2% at 24 h and 15% at 14 days, respectively, for both CRs evaluated.

Focusing on cell metabolic activity, a 48 h exposure to the diluted or undiluted extracts of all CRs tested did not significantly affect this aspect, except for the VD and UE resins (Figure 4). Unexpectedly, the use of an undiluted extract (100%) coming from the Bis-GMA-free VD resin was comparable to the effect of the Bis-GMA-based UE, since both had the most harmful effect on cell cultures among the different experimental groups. On the contrary, the effect on hGDF metabolic activity of another Bis-GMA-free material tested in this study (BF-2) and the Bis-GMA-based material (VOCO and ES-2) were comparable to the control condition. Those data support the concept that simply excluding the monomer Bis-GMA from CR formulation does not provide a steadily successful approach for reducing their cytotoxicity, since side effects can still occur due to the presence of other chemical components in the CR formulation. Indeed, multiple factors might concurrently contribute to the ultimate CR profile, whose assessment should be better accomplished by focusing on the whole material formulation instead of its individual components, thus opening the way for the in-depth investigation of such products and their effects.

To support this, a previous study reported that all the different components simultaneously present in the complex CR composition may induce effects that are different from what is expected by their action alone, as they could mutually influence one another through synergistic and/or antagonistic behaviors [22]. Therefore, an experimental plan based on the soaking of derived extracts, such as the one selected for this and for similar studies [2,38,39], appears to be the most suitable approach as it seems to better simulate a more clinically relevant scenario. However, future studies based on the approach of the direct contact between the CR disk and cells have been planned to confirm the role of dental resins at the cellular level.

Cell viability is not the only available outcome in the investigation of a potential cytotoxic effect towards cell cultures. The capability of resin monomers to influence cellular physiology and adaptive cell responses by increasing ROS production has already been reported by previous studies [33,47].

In this study, in accordance with cells metabolic activity results, ROS production induced by the exposure to undiluted extracts (Figure 5) was not even quantifiable in the VD and UE groups due to the extremely reduced number of living cells left after 48 h of treatment. However, it is important to underline that the cell treatments with the respective diluted extracts (50% and 25%) significantly increased fibroblast viability (Figure 5A,B), allowing for ROS quantification and suggesting a possible concentration-dependent mechanism, which could be used as a protective factor against their cytotoxic effects. Regarding BF-2, VOCO, and ES-2, a relatively low oxidative stress was recorded in these experimental groups since a ROS production comparable to the control was observed following the exposure to their respective 24-h and 14-day derived extracts. 

An additional way through which CRs may express their cytotoxic potential is represented by their capability to induce DNA damages such the DNA-Double Strand Breaks (DNA-DSBs) [48]. Therefore, the biological impact of our selected CRs was better investigated trough the evaluation of γH2AX expression, which is the early step in response to the induction of DNA-DSBs and its resulting damage [49].

As expected, immunofluorescence data confirmed that soaking extracts derived from BF-2, VOCO, and ES-2 were less genotoxic compared to VD and UE (Figure 6). Interestingly, despite VD being a Bis-GMA-free resin, it was the only resin characterized by a higher level of genotoxicity compared to the DOXO positive control at both times of soaking (24 h and 14 days), confirming its enhanced cytotoxic potential compared to the other CRs tested. 

It is important to underline that the present results are in line with what was observed by other research groups regarding the effect of CRs on oxidative stress [50] as well as cytotoxicity and genotoxicity [4,35,51]. Furthermore, we also confirmed data related to cell metabolic activity and ROS release by applying the same soaking-extracts approach to a murine fibroblast model (Balb/c-3T3 cells purchased from ATCC^®^, LGC Standards S.r.l.) (Appendix A). Indeed, VD and UE resins showed the highest cytotoxic potential, as far as the murine model was concerned and toward cells not belonging to the oral cavity, while BF-2, VOCO, and ES-2 again appeared as the most biocompatible resins (Appendix A).

Overall, these results confirm an extended potential cytotoxicity of CRs in vitro. However, we may suppose that the present findings, especially the worst cytotoxic effects herein observed, might be somewhat modulated following the clinical application of CRs in vivo, where oral cells and tissues can actively respond to monomer-induced stress with the activation of specific adaptive pathways. Therefore, further studies on the mechanisms underlying adaptive cell responses seem mandatory in order to improve the properties of dental restorative materials that encounter oral tissues and to develop effective strategies in dental therapy.

## 5. Conclusions

Enamel-BioFunction appeared definitely optimal in terms of cytotoxic effects. Similar findings were observed for Clearfil Majesty ES-2, despite the two resins showing different Bis-GMA/TEGDMA release patterns. Overall, our findings confirmed that the investigated CRs present cytotoxic and genotoxic potential, which do not only seem dependent on their actual Bis-GMA or TEGDMA release. Likewise, Enamel-plus HRi seemed notably more cytotoxic and genotoxic compared to Clearfil Majesty ES-2, notwithstanding their similar monomer releases. This suggests that simply excluding one specific and potentially cytotoxic monomer from CR formulation, such as Bis-GMA, might not steadily provide a successful approach for the improvement of their properties, which depends on the whole chemical composition of the material.

## Figures and Tables

**Figure 1 materials-14-05225-f001:**
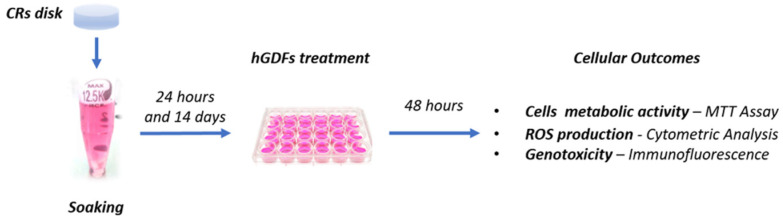
Experimental Plan. Soaking-derived extracts were obtained following the immersion of each type of CR disks of 50.27 mm^2^ surface in the culture medium (CTRL) for 24 h and 14 days (i.e., 2 CR disks in 1 mL of CTRL). The hGDFs isolated from human gingival biopsies were treated for 48 h with the culture medium (CTRL) or with the soaking extracts (24 h and 14 days) derived from the five selected CRs. Cell viability, ROS production, and genotoxicity were chosen as cellular outcomes.

**Figure 2 materials-14-05225-f002:**
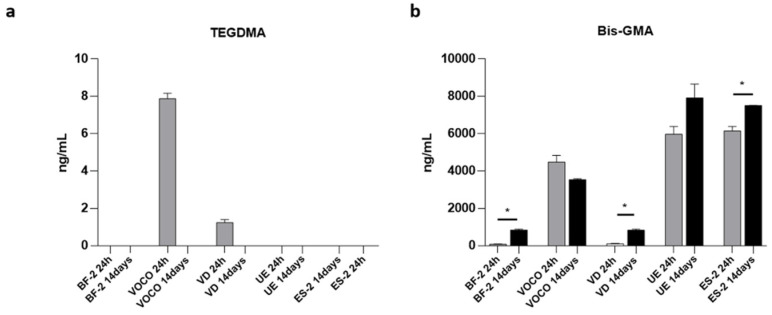
LC–MS/MS quantification. Levels of (**a**) TEGDMA and (**b**) Bis-GMA quantified in culture media following 24 h (in grey) and 14 days (in black) of CR disk immersion. Results are shown as mean ± standard error (SEM) (*n* = 3) (* *p* < 0.05 vs. 24 h).

**Figure 3 materials-14-05225-f003:**
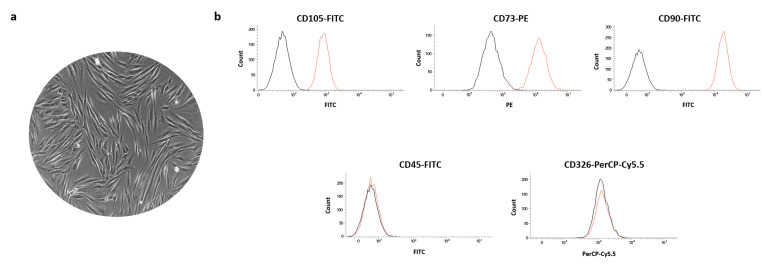
Characterization of hGDFs. (**a**) Representative phase contrast image of isolated hOBs. (**b**) Representative histograms of the CD105, CD73, CD90, CD45, and CD326 expressions evaluated through cytometric analysis.

**Figure 4 materials-14-05225-f004:**
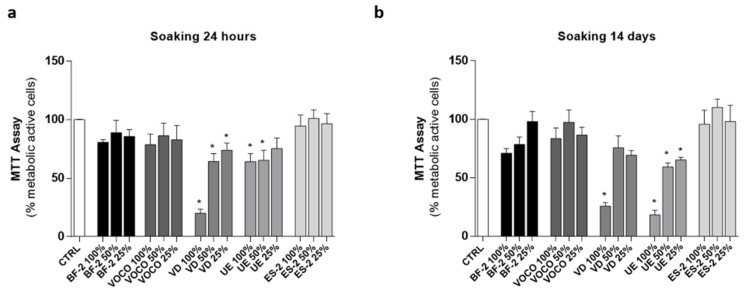
Cells metabolic activity. Effect of CRs on hGDF cell metabolic activity following 48 h of treatment with the (**a**) 24-h and (**b**) 14-day CR-derived extracts, undiluted (100%) or not (50% and 25%). Results are shown as mean ± standard error (SEM) (*n* ≥ 3) of the percentage (%) of metabolically active cells. (* *p* < 0.05 vs. CTRL).

**Figure 5 materials-14-05225-f005:**
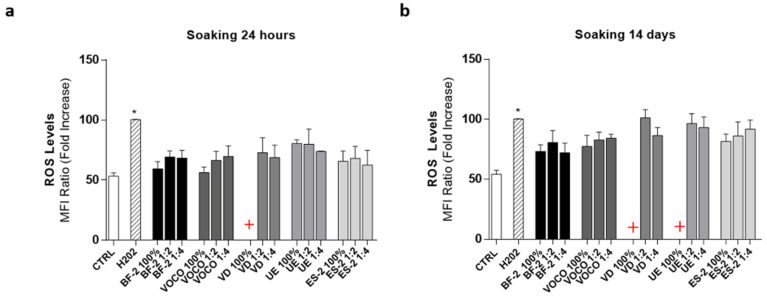
ROS levels. Cytometric analysis of ROS levels in hGDFs following 48 h of treatment with the (**a**) 24-h and (**b**) 14-day CR-derived extracts, undiluted (100%) or not (50% and 25%). Results, expressed as mean ± standard error (SEM) (*n* ≥ 3), are shown as the MFI ratio (fold increase) of ROS levels. (* *p* < 0.05 vs. CTRL). The red crosses indicate values not quantifiable (n.q.) in VD and UE groups due to the high level of cell mortality.

**Figure 6 materials-14-05225-f006:**
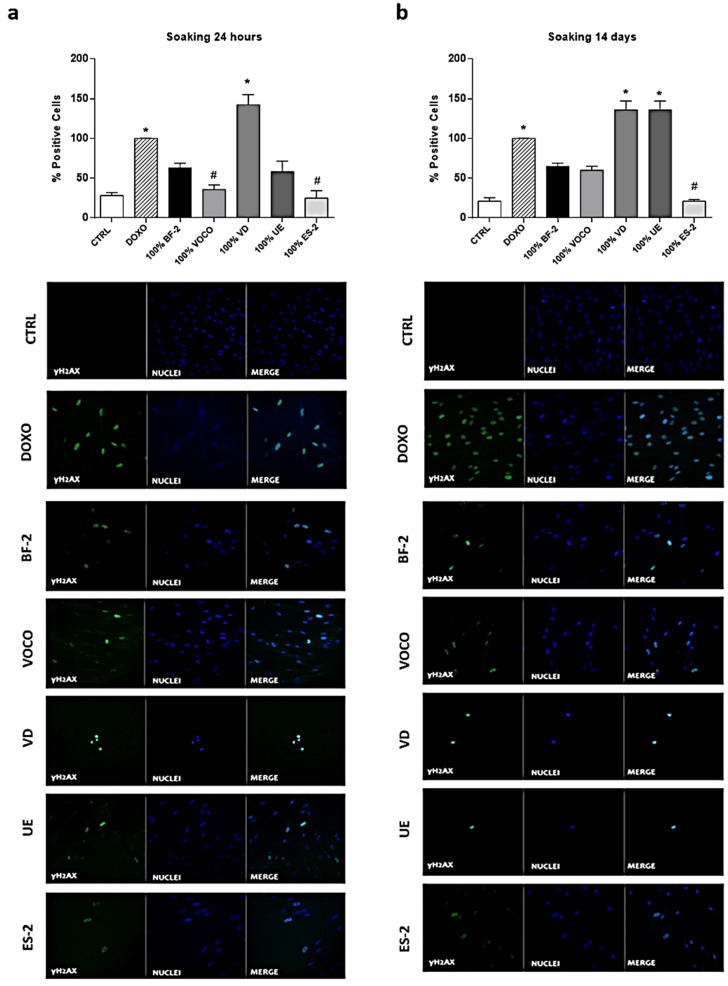
DNA damage. Graphs and representative fluorescence images show the expression of γH2AX in hGDFs (stained with anti-γH2AX, DAPI, and merged; 40x) following 48 h of treatment with the (**a**) 24-h and (**b**) 14-day CR-derived extracts, undiluted (100%). Results are expressed as mean ± standard error (SEM) (*n* ≥ 3). Data, calculated as percentage (%) of positive cells for γH2AX, were obtained by analyzing at least three different fields for each image with ImageJ software (NIH, USA). (* *p* < 0.05 vs. CTRL; # *p* < 0.05 vs. DOXO).

**Table 1 materials-14-05225-t001:** List and characteristics of the CRs included in the experimental design.

Group	Name	Shade	Manufacturer	Batch	Composition	Category
BF-2	Enamel BioFunction	BF2	Micerium S.p.A.	2019008149	74% wt fillers (5–50 nm silicon dioxide; 0,2–3 μm glass fillers)	Nanohybrid
			(Avegno, Italy)		UDMA, Tricyclodecane dimethanol dimethacrylate	
VOCO	GrandioSO	A2	Voco GmbH	1847313	89 % wt fillers (1 μm glass ceramic fillers; 20–40 nm silicon dioxide fillers)	Nanohybrid
			(Cuxhaven, Germany)		Bis-GMA, Bis-EMA, TEGDMA	
VD	Venus Diamond	A2	Kulzer GmbH	K010070	80–82% wt fillers (5 nm–20 μm barium aluminum fluoride glass fillers)	Nanohybrid
			(Hanau, Germany)		TCD-Urethaneacrylate, UDMA, TEGDMA	
UE	Enamel-plus HRi		Micerium S.p.A.		80% fillers wt (0.1 µm glass fillers, 20 nm zirconium nanoxides)	Nanohybrid
			(Avegno, GE, Italy)		UDMA, Bis-GMA, 1,4 Butanediol dimethacrylat (BDDMA)	
ES-2	Clearfil Majesty ES-2 Classic	A2	Kuraray	7D008	78% fillers wt (0.37 μm–1.5 μm silanated barium glass fillers, pre-polymerized organic fillers)	Nanohybrid
			(Chiyoda, Tokyo, Japan)		Bis-GMA, Hydrophobic aromatic dimethacrylate	

UDMA (urethane dimethacrylate); Bis-GMA (bisphenol-A-glycidyl-methacrylate); Bis-EMA (bisphenol-A-diglycidyl-methacrylate ethoxylated); TEGDMA (triethylene glycol dimethacrylate).

**Table 2 materials-14-05225-t002:** MRM transition, cone potential, and collision energy for each analyte.

Analyte	Parent (*m*/*z*)	Daughter (*m*/*z*)	Cone (V)	Collision Energy (eV)
TEGDMA	287.2	113.1	50	8
Bis-GMA	513.2	143.1	50	15

## Data Availability

Data available on request.

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
