# Peer review of "Cytotoxic and Genotoxic Effects of Composite Resins on Cultured Human Gingival Fibroblasts"

_materials, 2021, doi:10.3390/ma14185225_

Round 1

Reviewer 1 Report

It is an interesting work, but there are still some issues needed to be noticed.

  1. It is ineresting that Bis-GMA was detected in CRs of Bis-GMA free. Is it possile that you use the same mold to prepare the samples with and without Bis-GMA?
  2.  It should be noticed that all chosen CRs does not only have Bis-GMA and TEGDMA, but also have some other components, whether the difference in biocompatibility induced by the other components? It should be discussed. It is even better to test the release of some other components.
  3. All of Figures are not clear enough to be seen, please change by figures with higher resolution.
  4. Have you measured the DC of CRs? It could help you to discuss the results.

Author Response

We thank reviewer #1 for the appreciation of our manuscript as well as for his/her useful comments in response to which changes have been made. Particularly, the sections modified or added in the text are highlighted in red as track changes mode.

  • It is interesting that Bis-GMA was detected in CRs of Bis-GMA free. Is it possible that you use the same mold to prepare the samples with and without Bis-GMA?

We thank the reviewer for his/her observation. We confirm that different molds were used for different composites and, overall, great care was taken to avoid any possible contamination between different composites resins during sample preparation. In any case our results are consistent with data recently published by Šimková M set al  (Dental composites - a low-dose source of bisphenol A? Physiol Res. 2020) and, in agreement with this, it is important to point out that also in this study the release percentage of Bis-GMA from Bis-GMA-free resins is very low compared to the average release of other resins.

However, to satisfy the reviewer’s observation, we decided to better elucidate this specific issue in the discussion (see page 12).

  • It should be noticed that all chosen CRs does not only have Bis-GMA and TEGDMA, but also have some other components, whether the difference in biocompatibility induced by the other components? It should be discussed. It is even better to test the release of some other components.

 As reported in Table 1, we are aware that the CRs selected for this study not only have Bis-GMA and TEGDMA but also other methacrylate components, such as UDMA (Urethane Dimethacrylate). In addition different type of inorganic filler particles (zirconium, silica and barium) were also employed. Certainly, as suggested by the reviewer, all of these components can affect CRs biocompatibility. Nevertheless, in this study, we focused on the release evaluation of the most frequent metahacrylate monomers employed for dental CRs preparation such as TEGDMA and Bis-GMA, since several in vitro studies demonstrated their cytotoxic effects (see references 9-15). Of note, as in the case of the dental resin VenusDiamond (VD), our results suggest that, despite the removal of the Bis-GMA cytotoxic compound, biocompatibility can still be affected possibly due to the presence of other chemical components in CRs formulation. Thus, our data open the way for the in deep investigation of such materials and their biocompatibility. Therefore, following the reviewer’s suggestion, we added this concept in the discussion (see page 12).

  • All of Figures are not clear enough to be seen, please change by figures with higher resolution.

As suggested by the reviewer, we have replaced the all figures with the higher resolution ones (300 dpi) in the revised version of the manuscript.

  • Have you measured the DC of CRs? It could help you to discuss the results.

Thanks for this comment. As suggested by the reviewer, the final extent of monomer conversion might significantly influence the actual monomer release. This aspect could be particularly interesting, especially if the effects of different curing protocols had to be compared (Salehi S, Cytotoxicity of resin composites containing bioactive glass fillers. Dent Mater. 2015). However, in the present study, the curing protocol (light curing unit, irradiation time, irradiation distance, etc) was steadily the same for all samples/composites and it was not a variable under investigation. As a consequence, the Degree of Conversion (DC) was not measured: this was done according to similar previous studies that did not investigate the effect of different curing protocols too (Longo DL, Cytotoxicity and cytokine expression induced by silorane and methacrylate-based composite resins. J Appl Oral Sci. 2016; Shafiei F, Cytotoxic effect of silorane and methacrylate based composites on the human dental pulp stem cells and fibroblasts. Med Oral Patol Oral Cir Bucal. 2014; Krifka S, Oxidative stress and cytotoxicity generated by dental composites in human pulp cells. Clin Oral Investig. 2012).

Reviewer 2 Report

Thanks for the article and I have the following comments:

1. Many typos, like P5L168 H20 should be H2O , L167 A should be Å (Angstrom) , P4L125 mm2 should be mm2. Can you proof read all these details?

2. Please provide the representative spectra of LC-MS/MS for each analyte. Please also indicate the linear range (not only LOQ) of TEGDMA and bis-GMA

3. p4L129 "These two times of soaking were chosen in order to mimic acute and chronic monomers release from CRs." I could not agree. The method that you have used can only show how much residual monomers exist in the cured resin composites. It is not going to be released unless, at certain condition, you extract the monomers out. In your case, the methanol play a significant role. So, please delete this sentence, and/or you should be cautious to amend whole article for the proper wording should be "residual monomers", not the "released monomers"

4. The soaking details is not clear - your total surface area is 50.27mm2, ISO requirement is 100.54 mm2/ml, so how can you place the specimen in (plastic? glass? tube?) container without contacting any surface in 0.5ml medium?

5. What's the original TEGDMA and bis-GMA content in the resin composites before curing? Why the TEGDMA content was nil or less after 24 hrs but bis-GMA increased? Would your immersion media interact with the material? If you please this in water or PBS how would the results deviate? Please discuss.

6. Using the extract can only show the indirect biocompatibility. How about direct contact that should make more sense in resin composites? You should add this in the discussion.

Author Response

We thank reviewer #2 for the appreciation of our manuscript as well as for his/her useful comments in response to which changes have been made. Particularly, the sections modified or added in the text are highlighted in red as track changes mode.

  • Many typos, like P5L168 H20 should be H2O, L167 A should be Å (Angstrom), P4L125 mm2 should be mm2. Can you proof read all these details?

We have corrected the typos in the revised version.

  • Please provide the representative spectra of LC-MS/MS for each analyte. Please also indicate the linear range (not only LOQ) of TEGDMA and bis-GMA

In order to satisfy the request of the reviewer, we added a supplementary Figure (Figure Supplementary 1) showing the MS/MS spectrum and the Extracted Ion Chromatogram for each analyte. Moreover, we added the linear range of quantification and the r2 for TEGDMA and Bis-GMA in the material and method section 2.2 (see page 5).

  • p4L129 "These two times of soaking were chosen in order to mimic acute and chronic monomers release from CRs." I could not agree. The method that you have used can only show how much residual monomers exist in the cured resin composites. It is not going to be released unless, at certain condition, you extract the monomers out. In your case, the methanol play a significant role. So, please delete this sentence, and/or you should be cautious to amend whole article for the proper wording should be "residual monomers", not the "released monomers"

We thank the reviewer for his/her observation. As requested, in the revised version we have deleted the sentence “These two times of soaking were chosen in order to mimic acute and chronic monomers release from CRs”. In addition, we have replaced the wording “released monomers” with the proper one “residual monomer”.

  • The soaking details is not clear - your total surface area is 50.27mm2, ISO requirement is 100.54 mm2/ml, so how can you place the specimen in (plastic? glass? tube?) container without contacting any surface in 0.5ml medium?

As request by the reviewer, we have modified the paragraph relative to the soaking preparation (paragraph 2.1) in materials and methods section (see page 4) in order to better explain the soaking details.

  • What's the original TEGDMA and bis-GMA content in the resin composites before curing? Why the TEGDMA content was nil or less after 24 hrs but bis-GMA increased? Would your immersion media interact with the material? If you please this in water or PBS how would the results deviate? Please discuss.

Although the observation of the reviewer is absolutely plausible, we have to point out that the aim of the present study was to evaluate the biocompatibility of five different CRs toward a cellular model of primary isolated human gingival fibroblasts (hGDFs) by means of the dental composites soaking extracts approach. To this aim, the hGDFs were not place in a direct contact with the CRs disk but were treated with the soaking extracts derived by the immersion of CRs disk in the culture medium for 24 hours and 14 days. Therefore, we did not evaluate the original TEGDMA and bis-GMA content in the resin composites before curing but we assessed the quantifiable Bis-GMA and TEGDMA residual monomer in the soaking extract by LC-MS/MS analysis.

In addition, we are aware that the TEGDMA content was nil or less after 24 hours but Bis-GMA increased. As previously reported by Moharamzadeh K et al (HPLC analysis of components released from dental composites with different resin compositions using different ex-traction media. J Mater Sci Mater Med. 2007), the authors demonstrated that TEGDMA monomers disappear after seven days of incubation of TEGDMA-based CRs in culture media due to their enzymatic degradation and aggregation mainly with albumin. Our data are in agreement with this phenomenon showing undetectable levels of TEGDMA monomers after 14 days of exposure.

Moreover, several other studies (Finer Y., Biodegradation of a dental composite by esterases: dependence on enzyme concentration and specificity. Journal of Biomaterials Science, Polymer Edition 2003; Shajii L., Effect of filler content on the profile of released biodegradation products in micro-filled bis-GMA/TEGDMA dental composite resins. Biomaterials. 1999) have largely demonstrated that soaking media directly affects the integrity of CRs and their monomers release: specifically, an enzyme-catalysed hydrolysis of CRs can occur by esterases, as those typically present in inflammatory sites and saliva. Since our work was carried out to investigate the effects of CRs-released monomers on gingival fibroblast cell models for in vitro toxicity studies, the quantification of TEGDMA and Bis-GMA was performed on specimens deriving from the soaking of CRs into culture media with serum, thus comprising enzymes (such as esterases) among the other components. Accordingly, both the experience by Finer and Santerre of decreasing amount of residual TEGDMA monomer by esterases when  compared to PBS soaking and the awareness of the complexity and enzymatic heterogeneity of the oral cavity where CRs are supposed to be placed raised the importance to mimic the in vivo model by conceiving the idea that serum could catalyse the rate of hydrolysis in methacrylate-based polymers, as it would probably occur in the oral cavity depending on the individual’s saliva enzymes make-up. On the other hand, regarding the  increasing levels of Bis-GMA, there is a general consensus that esterases do not show the same specificity for monomers and that CRs degradation relies on the specific enzyme dose-response (Santerre JP, Relation of dental composite formulations to their degradation and the release of hydrolyzed polymeric-resin-derived products. Crit Rev Oral Biol Med. 2001). In order to answer the questions raised by the reviewer, we enriched the discussion by briefly adding the concepts described and the new references reported above (see page 11).

  • Using the extract can only show the indirect biocompatibility. How about direct contact that should make more sense in resin composites? You should add this in the discussion

We thank the reviewer for his/her comment since suggests to plan future interesting studies. Indeed, although the experimental approach mainly applied at the moment is based on the cellular treatment with the dental composites soaking derived-extracts (since it appears the most suitable approach as it seems to better simulate a more clinically relevant scenario) we concur with the reviewer observation that it would be interesting additionally investigate the biocompatibility effect derived from the direct CRs contact with the cells, possibly dental pulp derived cells. Therefore, as suggested by the reviewer, we better explain our experimental choice (see page 12 and additional recent references 38-39: Bandarra S, Biocompatibility of self-adhesive resin cement with fibroblast cells. J Prosthet Dent. 2021; Landenberger P et al., The effect of new anti-adhesive and antibacterial dental resin filling materials on gingival fibroblasts, Dent Mater 2021) and we add the concept of a possible experimental future plan in the discussion (see page 12).

Reviewer 3 Report

The authors present an in vitro study focusing on the cytotoxic effects of five different composite resins on human gingival fibroblasts. The manuscript introduction provides the relevant information, and the performed experiments are appropriate for the study aim and well-conducted. However, I have some suggestions to improve the manuscript quality.

The authors use the term biocompatibility several times, which should be replaced by cytotoxicity (or its absence). As the authors define in the introduction section, biocompatibility is more appropriate for in vivo and clinical studies since it refers to contact with living tissues.

Table 1: please provide the appropriate caption for the table (UDMA, Bis-GMA, Bis-EMA, and TEGDMA)

Lines 124-127: I believe the authors refer to the ISO 10993-12 when referring to the material/extraction medium ratio. How was the 100.54 mm2/mL determined? It was considered a material with a thickness > 1mm, as referred to in the ISO 10993-12, which corresponds to a 1.25cm2/mL ratio?

During the soaking procedure, were the containers static or in rotation?

Line 128: what kind of control medium was used for extraction? DMEM? Please clarify.

Were the conditioned mediums filtered, centrifugated, or processed before use?

Why were the tests performed after 48 hours? How was this time chosen?

Lines 207-208: the authors refer to 100% concentration and after to 1:2 and 1:4. I suggest referring to 50% and 25% concentrations to unify the terminology.

Line 214: please refer to cellular concentration in cells/ml instead of cell/well, since this will not allow the experiment replication because the plate type is not indicated.

Section 2.6: which reactive oxygen species does the kit used evaluates?

Section 2.8: was the data normality/not normality evaluated before choosing the statistical test?

Figure 2 presents relevant data from the TEGDMA and Bis-GMA release quantification. However, I found the discussion about these values insufficient. Are they clinically relevant? Are they above or below the values clinical reported for release from different materials? See, for instance, https://doi.org/10.3390/ijerph16091627. Please discuss this.

I do not agree with the term “cell viability” when referring to the MTT assay. The MTT assay evaluates the cell’s metabolic activity and is not a cell viability test. Of course, it can give an indirect measure of viability, but cells can be viable but senescent, for instance, and no metabolic activity is observed. For instance, appropriate tests for cell viability are trypan blue exclusion assay or labeling with AnV/IP. Please correct it. 

Figure 5: the authors refer to “CTRL” that should be referred to as negative control since H2O2 is also a control, but a positive one

Figure 6: the authors refer to “CTRL,” which should be referred to as negative control since DOXO is also a control but a positive one.

Lines 366-377: I suggest this information to be moved to the introduction section.

Lines 407-412: do authors have a possible explanation for these results? What other component of the Bis-GMA-free resin could have induced the metabolic activity decrease?

Line 423- 425: I agree with the authors that testing the material instead of isolated compounds is preferable. But why choose the indirect contact methodology with extracts instead of direct contact? Please discuss this.

Line 435: I suggest replacing the term “dilution dependent mechanism” with “concentration-dependent”, since what dilution does is decreasing the material´s concentration.

I found the obtained results interesting, but in my opinion, the discussion lacks a more comprehensive interpretation. For example, do authors think these results are similar in the clinical situation, where defense mechanisms, as antioxidants defenses, for instance, exist? Please discuss it.

Author Response

We thank reviewer #3 for the appreciation of our manuscript as well as for his/her useful comments in response to which changes have been made. Particularly, the sections modified or added in the text are highlighted in red as track changes mode.

  • The authors use the term biocompatibility several times, which should be replaced by cytotoxicity (or its absence). As the authors define in the introduction section, biocompatibility is more appropriate for in vivoand clinical studies since it refers to contact with living tissues.

We thank the reviewer for his/her observation. As suggested we replaced the term biocompatibility with “cytotoxicity”.

  • Table 1: please provide the appropriate caption for the table (UDMA, Bis-GMA, Bis-EMA, and TEGDMA)

As request by the reviewer, we have provided the appropriate capitation for UDMA, Bis-GMA, Bis-EMA, and TEGDMA in the table 1.

  • Lines 124-127: I believe the authors refer to the ISO 10993-12 when referring to the material/extraction medium ratio. How was the 100.54 mm2/mL determined? It was considered a material with a thickness > 1mm, as referred to in the ISO 10993-12, which corresponds to a 1.25cm2/mL ratio?

Thanks for this comment. Based on previous studies, these authors found a certain heterogeneity concerning the material/medium ratios. Krifka et al. used a 0.916 cm2/mL ratio (Krifka S, Oxidative stress and cytotoxicity generated by dental composites in human pulp cells. Clin Oral Investig. 2012); Shafiei et al. employed a 1.7 cm2/mL ratio (Shafiei F, Cytotoxic effect of silorane and methacrylate based composites on the human dental pulp stem cells and fibroblasts. Med Oral Patol Oral Cir Bucal. 2014); Salehi et al. adopted a 2.12 cm2/mL ratio (Salehi S, Cytotoxicity of resin composites containing bioactive glass fillers. Dent Mater. 2015). Moreover, Shafiei et al. indicated, for the material/medium ratio, a “recommended range” varying between 0.5 cm2/mL and 6.0 cm2/mL, suggesting that a certain deviation from the standard might be considered acceptable, as long as the actual ratios used are carefully reported. On the above bases and in order to ease the experimental procedures, in the present study, for every different extract, two cylindrical discs (having each one a total surface of 50.27 mm2) were placed inside a plastic tube containing 1.0 mL of culture medium, leading to a 1.0054 cm2/mL ratio (or 100.54 mm2/mL), which is not extremely far from the 1.25 cm2/mL ratio that has been properly indicated by the reviewer. In accepting the reviewer observation, paragraph 2.1 (page 4) was slightly rephrased, in order to avoid the misunderstanding that the 1.0054 cm2/mL material/medium ratio strictly matches the ISO requirements.

  • During the soaking procedure, were the containers static or in rotation?

We thank the reviewer for the observation permitting us a better explanation of the experimental procedures. During the soaking procedure tubes, containing the CRs disks and culture medium, were placed in incubation in static condition at 37 °C for 24 hour and 14 days. However, one a day we have shaken tubes manually. Based on the reviewer observation, we added this in the revised manuscript (see page 4 paragraph 2.1).

  • Line 128: what kind of control medium was used for extraction? DMEM? Please clarify.

As request by the reviewer, we have modified the paragraph relative to the soaking preparation (paragraph 2.1) in materials and methods section (see page 4 paragraph 2.1) in order to better explain the soaking details.

  • Were the conditioned mediums filtered, centrifugated, or processed before use?

The CRs’ soaking derived-extracts were not filtered, centrifuged or processed, the hGDFs were treated with the soakings as it is (100%) or diluted (50% and 25%).

  • Why were the tests performed after 48 hours? How was this time chosen?

We thank the reviewer for his/her observation as it gives us the opportunity to better explain our choice. Previous published studies (Longo DL, Cytotoxicity and cytokine expression induced by silorane and methacrylate-based composite resins. J Appl Oral Sci. 2016; Krifka S, Oxidative stress and cytotoxicity generated by dental composites in human pulp cells. Clin Oral Investig. 2012; Salehi S, Cytotoxicity of resin composites containing bioactive glass fillers. Dent Mater. 2015) showed that cells were treated for 24 hours with the resin derived-extracts. Based on these, we decided to test a longer time of cell exposure to the CRs soaking extracts (48 hours) in order to investigate the effects of CRs on more complex cellular pathways such as DNA damage (genotoxicity). Furthermore, in order to standardize the treatment time, we chosen 48 hours of treatment also for the experiments regarding cell metabolic activity and ROS production.

  • Lines 207-208: the authors refer to 100% concentration and after to 1:2 and 1:4. I suggest referring to 50% and 25% concentrations to unify the terminology.

Following the reviewer’s suggestion, we used in the revised version of the manuscript the terminology 50% and 25% concentrations.

  • Line 214: please refer to cellular concentration in cells/ml instead of cell/well, since this will not allow the experiment replication because the plate type is not indicated.

In order to satisfy the request of the reviewer, in the revised version we have referred to cellular concentration in term of cells/ml instead of cell/well. In addition we also specify the wells used in the experiments to allow their replication.

  • Section 2.6: which reactive oxygen species does the kit used evaluates?

As reported in the paragraph 2.6, we used CellROXTM Green Reagent” commercially kit (C10444, Molecular Probes, Eugene) for the quantification of ROS release. CellROX® Green Reagent is a novel fluorogenic probe for measuring oxidative stress in live cells. Although the reviewer request is correct, unfortunately, the manufacturer’s instruction does not which reactive oxygen species is specifically evaluated.

  • Section 2.8: was the data normality/not normality evaluated before choosing the statistical test?

Thanks for this comment. The data were previously analyzed by using the Shapiro-Wilk normality test before choosing the proper statistical test. Therefore, we have added this in the revised version (see page 7 paragraph 2.8).

  • Figure 2 presents relevant data from the TEGDMA and Bis-GMA release quantification. However, I found the discussion about these values insufficient. Are they clinically relevant? Are they above or below the values clinical reported for release from different materials? See, for instance, https://doi.org/10.3390/ijerph16091627. Please discuss this.

Thanks for the valuable comment. In the present study, the LC-MS/MS quantification test showed a BIS-GMA release ranging between 8000 ng/mL (for UE on the 14-days extracts) and 79.7 ng/mL (for BF-2 on the 24-h extracts). Based on the review suggested by the reviewer (Paula AB, Once Resin Composites and Dental Sealants Release Bisphenol-A, How Might This Affect Our Clinical Management?-A Systematic Review. Int J Environ Res Public Health. 2019), those results might be considered clinically relevant, as their magnitude appears close to the maximum levels of BIS-GMA clinically quantified in human saliva (2149 ng/mL) by Michelsen et al. (Michelsen, V.B Detection and quantification of monomers in unstimulated whole saliva after treatment with resin-based composite fillings in vivo. Eur. J. Oral Sci. 2012) after having exposed patients to another BIS-GMA based composite (Filtek Z250).

On the other hand, TEGDMA levels were generally lower compared to BIS-GMA, ranging in the present in vitro study between 7.9 ng/mL (for VOCO on the 24-h extracts) and 0 ng/mL. Similarly, TEGDMA resulted just clinically detectable but not quantifiable in human saliva by Michelsen. Therefore, this has been addressed to the reader in the discussion section (page 12).

  • I do not agree with the term “cell viability” when referring to the MTT assay. The MTT assay evaluates the cell’s metabolic activity and is not a cell viability test. Of course, it can give an indirect measure of viability, but cells can be viable but senescent, for instance, and no metabolic activity is observed. For instance, appropriate tests for cell viability are trypan blue exclusion assay or labeling with AnV/IP. Please correct it. 

As properly suggested by the reviewer, in the revised version we modified the term “cell viability” with “Cell metabolic activity”.

  • Figure 5: the authors refer to “CTRL” that should be referred to as negative control since H2O2 is also a control, but a positive one

We thank the reviewer for his/her observation because we can following clarify that the CTRL condition refers to hGDFs treated with their growth culture medium, therefore it is a basal condition. In the revised version we have elucidate this aspect (see pages 8 and 9).

  • Figure 6: the authors refer to “CTRL,” which should be referred to as negative control since DOXO is also a control but a positive one

As for the previous comment, also in this case the CTRL condition refers to hGDFs treated with their growth culture medium, therefore it is a basal condition.

  • Lines 366-377: I suggest this information to be moved to the introduction section.

Following the reviewer suggestion, in the revised version we have improved the introduction section with the concepts explained between the lines 366-377 (see page 2).

  • Lines 407-412: do authors have a possible explanation for these results? What other component of the Bis-GMA-free resin could have induced the metabolic activity decrease?

           As reported in Table 1, the CRs selected for this study not only have Bis-GMA but also other metahacrylate components, such as TEGDMA and UDMA (Urethane Dimethacrylate). In addition different type of inorganic filler particles (zirconium, silica and barium) were also employed. Certainly, all of these components could affect CRs metabolic activity. Nevertheless, in this study, we focused on the release evaluation of the most frequent metahacrylate monomers employed for dental CRs preparation such as TEGDMA and Bis-GMA, since several in vitro studies demonstrated their cytotoxic effects (see references 9-15). Of note, as in the case of the dental resin VenusDiamond (VD), our results confirm that, despite the removal of the Bis-GMA cytotoxic compound, biocompatibility can still be affected possibly due to the presence of other chemical components in CRs formulation. Thus, our data open the way for the in deep investigation of such materials and their biocompatibility. Therefore, following the reviewer’s suggestion, we added this concept in the discussion (see page 12)

  • Line 423- 425: I agree with the authors that testing the material instead of isolated compounds is preferable. But why choose the indirect contact methodology with extracts instead of direct contact? Please discuss this.

We thank the reviewer for his/her comment since suggests to plan future interesting studies. Indeed, although the experimental approach mainly applied at the moment is based on the cellular treatment with the dental composites soaking derived-extracts (since it appears the most suitable approach as it seems to better simulate a more clinically relevant scenario) we concur with the reviewer observation that it would be interesting additionally investigate the biocompatibility effect derived from the direct CRs contact with the cells, possibly dental pulp derived cells.

Therefore, as suggested by the reviewer, we better explain our experimental choice (see page 12 and additional recent references 38-39: Bandarra S, Biocompatibility of self-adhesive resin cement with fibroblast cells. J Prosthet Dent. 2021; Landenberger P et al., The effect of new anti-adhesive and antibacterial dental resin filling materials on gingival fibroblasts, Dent Mater 2021) and we add the concept of a possible experimental future plan in the discussion (see page 12).

  • Line 435: I suggest replacing the term “dilution dependent mechanism” with “concentration-dependent”, since what dilution does is decreasing the material´s concentration.

As required by the reviewer, in the revised version we have replaced the term “dilution dependent mechanism” with “concentration-dependent”.

  • I found the obtained results interesting, but in my opinion, the discussion lacks a more comprehensive interpretation. For example, do authors think these results are similar in the clinical situation, where defense mechanisms, as antioxidants defenses, for instance, exist? Please discuss it.

We thank the reviewer for his/her proper observation. In vivo, human gingival fibroblasts are among the first cells to be in contact with substances eluted from dental materials such as resins composites. Previous studies, according to the present research, clearly demonstrated that the substances extracted from resin monomers (such as Bis-GMA, TEGDMA, HEMA and so on) specifically interfere in vitro with important cellular functions inducing cytotoxicity, oxidative stress and DNA strand breaks (Krifka S, Oxidative stress and cytotoxicity generated by dental composites in human pulp cells. Clin Oral Investig. 2012; Yang Y, Cytotoxicity and DNA double-strand breaks in human gingival fibroblasts exposed to eluates of dental composites. Dent Mater. 2018). Concerning the clinical situation, it is well known that residual monomers may act also in vivo as environmental stressors, which inevitably disturb regulatory cellular networks by interfering with specific signal transduction pathways. However, although the residual monomer concentrations found in vitro might be considered comparable to those clinically quantified in human saliva (as underlined in a previous answer), in vivo studies showed the amounts of monomers clinically released from dental composites to be far below the levels required to induce systemic adverse effects (Seiss M, Quantitative determination of TEGDMA, BHT, and DMABEE in eluates from polymerized resin-based dental restorative materials by use of GC/MS. Arch Toxicol 2009). From this point of view, as suggested by the reviewer, it is important to underline that, in vivo, oral cells and tissues may actively respond to monomer-induced stress with the activation of specific adaptive pathways (i.e. expression of antioxidants enzyme, activation of signaling proteins leading to the regulation of the cell cycle) (Krifka S, A review of adaptive mechanisms in cell responses towards oxidative stress caused by dental resin monomers. Biomaterials. 2013). Based on these evidence, to reply to reviewer’s acute observation, we may confidently suppose that our in vitro findings, especially the strong cytotoxic effects induced by the commercially available VenusDiamond, might be somehow modulated following the CR application in vivo. Therefore, we added this concept in the revised version (see page 13).

Round 2

Reviewer 1 Report

It is obvious that the author has answered the questions, and can be accepted now.

Author Response

Thanks to the reviewer for his/her helpful comments

Reviewer 2 Report

I appreciate the authors have added sufficient information and now this manuscript has a good standard to publish, subject to some very minor typo such as L160 H20 --> H2O.

Author Response

Thanks to the reviewer for his/her helpful comments. As indicated, we have corrected the typos in the text.